# A Label and Probe-Free Zika Virus Immunosensor Prussian Blue@carbon Nanotube-Based for Amperometric Detection of the NS2B Protein

**DOI:** 10.3390/bios11050157

**Published:** 2021-05-16

**Authors:** Bárbara V. M. Silva, Marli T. Cordeiro, Marco A. B. Rodrigues, Ernesto T. A. Marques, Rosa F. Dutra

**Affiliations:** 1Biomedical Engineering Laboratory, Department of Biomedical Engineering, Federal University of Pernambuco, Recife 50670-901, Brazil; barbaravmsilva@hotmail.com; 2Department of Virology and Experimental Therapy, Oswaldo Cruz Foundation—FIOCRUZ, Aggeu Magalhães Institute, Recife 50670-420, Brazil; marli@cpqam.fiocruz.br; 3Department of Electronics and Systems, Federal University of Pernambuco, Recife 50670-901, Brazil; marco.rodrigues@ufpe.br; 4Center for Vaccine Research, Department of Infectious Diseases and Microbiology, University of Pittsburgh, Pittsburgh, PA 15213, USA; marques@pitt.edu

**Keywords:** Zika virus, NS2B, Prussian blue, point-of-care testing, probeless, reagentless

## Abstract

Zika virus (ZIKV) is a mosquito-borne infection, predominant in tropical and subtropical regions causing international concern due to the ZIKV disease having been associated with congenital disabilities, especially microcephaly and other congenital abnormalities in the fetus and newborns. Development of strategies that minimize the devastating impact by monitoring and preventing ZIKV transmission through sexual intercourse, especially in pregnant women, since no vaccine is yet available for the prevention or treatment, is critically important. ZIKV infection is generally asymptomatic and cross-reactivity with dengue virus (DENV) is a global concern. An innovative screen-printed electrode (SPE) was developed for amperometric detection of the non-structural protein (NS2B) of ZIKV by exploring the intrinsic redox catalytic activity of Prussian blue (PB), incorporated into a carbon nanotube–polypyrrole composite. Thus, this immunosensor has the advantage of electrochemical detection without adding any redox-probe solution (probe-less detection), allowing a point-of-care diagnosis. It was responsive to serum samples of only ZIKV positive patients and non-responsive to negative ZIKV patients, even if the sample was DENV positive, indicating a possible differential diagnosis between them by NS2B. All samples used here were confirmed by CDC protocols, and immunosensor responses were also checked in the supernatant of C6/36 and in Vero cell cultures infected with ZIKV.

## 1. Introduction

In response to the unprecedented ZIKV outbreak, an emerging mosquito-borne flavivirus in the Americas, the World Health Organization (WHO) was prompted to declare ZIKV a “Public Health Emergency of International Concern” in February 2016 [1]. Brazil was the most affected country showing a dramatic increase in reports of ZIKV infection in the Americas, with ≈95% of all cases of suspected ZIKV-associated congenital disease [2,3]. ZIKV is associated with fetal microcephaly in at least one in ten pregnancies, intrauterine growth retardation, and other congenital malformations [4], and has also been associated with the Guillain-Barré syndrome in adults [5]. However, long-term costs will likely be much higher given the additional, as-yet-unknown complications from congenital infections.

The molecular diagnosis of ZIKV infection based on reverse transcription quantitative real-time polymerase chain reaction (RT-qPCR) is considered the gold-standard detection method for active infections [6,7]. Although some molecular biosensors have been recently produced for ZIKV [8,9], including point-of-care testing, the efficacy of RNA detection is limited to the first few weeks after the onset of the virus infection [10]. Regarding ZIKV serological diagnosis, its accuracy for IgM or IgG antibody detections is limited due to cross-reactivity with other flaviviruses, especially dengue (approximately 55.6% homology) [11,12]. Many researchers have been challenged to develop new diagnostic methods and biomarkers in order to avoid the cross-reactivity with dengue, yellow fever, West Nile, Oropouche, and Mayaro viruses [13]. Recently, Mishra et al. developed an enzyme-linked immunosorbent assay (ELISA) using the non-structural protein NS2B as a powerful biomarker for ZIKV, demonstrating a high sensitivity (96%) in ZIKV in the early convalescent-phase [14]. NS2B (14 KDa) is essential for cleavage of the polyprotein precursor ZIKV and the generation of fully functional viral proteins [15]. In addition, NS2B presents a low genomic similarity when compared with others flaviviruses [16]. Given the need for mitigating the effects of ZIKV, rapid and practical point-of-care testing was developed.

Screen-printed electrodes (SPE) using carbon ink have been widely used for mass-production due to their low cost, diversity of printing on different substrates, including flexible plastic and ceramics, and miniaturized size, providing easy integration with electronic circuits [17]. Carbon nanotubes and other carbon allotrope nanomaterials have been used to improve the SPE performance due to their high surface areas (surface/volume ratio), high conductivity and easy chemical functionalization [18,19]. Regarding electrochemical immunosensor approaches, label-free detection of ZIKV using voltammetry and impedance spectroscopy techniques have been proposed, and the antigen–antibody interactions are measured at the diffusion barrier by decreasing the transfer of electroactive species released by adding redox probe solutions [20,21,22]. This additional and inconvenient step has been stated as one limitation of this technique in point-of-care. In this sense, our group has proposed new redox probe platforms within which electroactive species are supplied [23,24,25]. Catalytic nanocomposites can act as a redox inherent effective mediator, allowing the electroactive species to come from their surface to the electrode, avoiding the conventional addition of redox probe solutions. Among the electroactive mediators, Prussian blue (PB), as a well-known “artificial peroxidase”, has been studied extensively due to its high catalytic activity and excellent reversibility in electrochemical applications [21]. Conductive polymers in association with PB have shown a successful utility, facilitating the interfacial electron transfer and simultaneously improving the stability of the PB [22,23]. Polypyrrole (PPy) is one of the most commonly used conductive polymers for electrochemical sensors due to its excellent reducibility, easy synthesis and high conductivity [24]. In order to make use of these advantages, a novel screen-printed immunosensor making use of CNTs functionalized with PB integrated in the polymeric matrix was proposed for direct electrochemical detection of the NS2B protein, a potential marker of acute phase ZIKV, which is one of most severe mosquito-borne infections and a self-limiting disease.

## 2. Materials and Methods

### 2.1. Reagents

Py (98%), multi-walled carbon nanotube functionalized with carboxylic acid (COOH-CNT) (>8% functionalized), glycine, potassium ferricyanide (K_3_[Fe(CN)_6_]) and iron (III) chloride hexahydrate (FeCl_3_^.^6H_2_O) were obtained from Sigma-Aldrich (St. Louis, MO, USA). Polyclonal antibody to Zika virus (ZIKV) against NS2B protein produced in rabbit was obtained from GeneTex (Irvine, CA, USA). Graphite powder acquired from Fluka (St. Louis, USA) and carbon ink (Electrodag PF-407C) purchased from Acheson (Port Huron, MI, USA) were used to manufacture the screen-printed carbon electrode (SPE). Phosphate buffer saline (PBS) (0.01 mol L^−1^, pH 7.4) was used in all experiments for dilution of the samples. Ultrapure water obtained from a Millipore water purification system (18 MΩ, Milli-Q, Millipore) was utilized in all assays. All chemicals were of analytical grade.

### 2.2. ZIKV Isolates Culture and Serum Samples

The ZIKV isolates were inoculated on Vero cells containing Dulbecco’s Modified Eagle’s medium (DMEM). The viral suspension shows a titer of 2.2 × 10^6^ PFU^.^mL^−1^ at a multiplicity of infection of 0.1 for 4–5 days. Viral stocks were then produced in Vero cells and stored at −80 °C until use. Samples of the culture cells without ZIKV isolates were used as control study. To obtain the analytical curve, the supernatants of ZIKV isolate cultures were diluted in PBS.

The serum samples were from patients admitted to a clinical hospital (Pernambuco, Brazil) with molecular and serological diagnosis confirming ZIKV. The ZIKV positive serum was defined as a RT-PCR positive titer from ZIKV and positive from IgM ELISA assay according to CDC protocols. All controls used here (ZIKV negative) were positive for dengue virus infections in order to study the cross-reactive immunosensor. These serum controls were defined as RT-PCR titer from negative ZIKV, but positive from DENV and IgM and IgG ELISA assay DENV, and negative against ZIKV. NS2B presence was confirmed by in-house ELISA assay, which was developed by Prof. Ernesto Marques’s group against antibody anti-NS2B acquired by Genetex (USA). All patients were admitted to this research according to ethical procedures established by the Aggeu Magalhes Human Research committee (protocol number 63441516.6.0000.5190).

### 2.3. Apparatus

Electrochemical studies were performed using an Autolab PGSTAT204 analysis system with Nova 2.1.1 software from Eco Chemie (Utrecht, NLD). All measurements were made using a three-electrode system comprised of SPE as the working electrode, a helical platinum wire as an auxiliary electrode, and an Ag/AgCl (KCl _sat_) electrode as reference. All potentials given in this work were determined relative to the Ag/AgCl (KCl _sat_) reference electrode.

Chemical characterization of the PB@CNT-PPy nanocomposite was performed by Fourier transform infrared (FT-IR) spectroscopy analyses. FT-IR spectra were recorded on a Bruker FT-IR spectrometer, Model IFS-66 (Ettlingen, DEU, Germany), controlled by OPUS software (version 6.5). The measurements were made in the attenuated total reflectance (ATR) mode with a frequency resolution of 25 cm^−1^ at room temperature (24 °C) and controlled humidity (~10%).

### 2.4. SPE Manufacture

The electrode composition (*w*/*w*) included a carbon ink modified with graphite powder (10%), which was mixed together for 15 min and then squeezed on a plastic mask over rectangular polyethylene terephthalate surface to obtain a thin film, according to Silva et al. (2016). The mask contained a circular working area connected to a rectangular area (2.0 mm × 10.0 mm) that was used as the electrical contact. The active area of the working electrode (4.0 mm in diameter) was delimited by galvanoplasty tape (3M). Then, after printing the carbon ink, the electrode was dried at 60 °C for 20 min. Prior to use, the surface of the SPE was carefully polished for 2 min and then submitted to an electrochemical pre-treatment by recording 40 cyclic voltammograms between 2.0 V and −2.0 V (vs. Ag/AgCl (KCl _sat_)) at 0.1 Vs^−1^ in a KCl solution (0.1 mol L^−1^) [23].

### 2.5. Preparation of the PB@CNT-PPy Nanocomposite

For obtaining the nanocomposite film, COOH-CNT was dispersed in a solution of the FeCl_3_^.^H_2_O (0.2 mol L^−1^) and prepared in dimethylformamide in an ultrasonic bath for 2 h. The cleaned SPE surface was modified with 3 µL of the COOH-CNT (1 g L^−1^) prepared in the FeCl_3_ solution. The modified electrode was kept at 50 °C for 15 min in order to evaporate the solvent. Afterwards, the modified electrode was immersed in an electrochemical cell containing Py monomer (0.1 mol L^−1^) and K_3_[Fe(CN)_6_] (0.01 mol L^−1^) prepared in a KCl solution (0.1 mol L^−1^). The nanocomposite film was synthesized by submitting the SPE to 20 voltammetric cycles in the potential range from 0.8 to −0.8 V (vs. Ag/AgCl (KCl _sat_)) with a scan rate of 0.02 V s^−1^. The as-prepared electrode was named PB@CNT-PPy/SPE.

### 2.6. Anti-NS2B Immobilization

The residual carboxylic groups of PB@CNT-PPy on the SPE surface were activated with EDC (0.02 mol L^−1^) and NHS (0.05 mol L^−1^) solution for covalent immobilization of the anti-NS2B. Then, an aliquot of the anti-NS2B solution (0.025 g L^−1^) was dropped on the electrode surface and kept for 1 h in a moist chamber at 4 °C. The non-specific bindings and remaining activated carboxylic groups were blocked for 30 min with glycine solution (0.1 mol L^−1^, pH 9.0). All assembling steps of the immunosensor were monitored using cyclic voltammograms (CVs) in the potential range from −0.8 to 1.0 V (vs. Ag/AgCl (KCl _sat_)) at 0.05 V s^−1^ in 0.1 mol L^−1^ KCl.

### 2.7. Analytical Measurements

The analytical responses of the immunosensor were performed by changing the current after incubation with ZIKV supernatant from cell-culture samples (after seven days of cultivation), and serum samples of patients. The analytical responses of antigen–antibody reactions were obtained by chronoamperometry at a fixed potential of 0.4 V for 10 s, and the current response curve was registered at 2 s after applying the potential. Before applying potential of 0.4 V, the sensor was established at 0 V for 25 s. Responses of NS2B binding were obtained through current values after sample incubations subtracting from the blank (before incubating) using the following equation:I% = (I_BLANK_ − I_SAMPLE_)/I_BLANK_) × 100(1)
where I_BLACK_ is the current value of the as-ready immunosensor and I_SAMPLE_ is the current value after exposure to viral ZIKV culture or serum samples. All preparation steps performed on the ZIKV immunosensor and chronoamperometric measurements are represented in Figure 1.

## 3. Results and Discussion

### 3.1. Characterization of PB@CNT-PPy Nanocomposite

The electrosynthesis of the PB@CNT-PPy compound employing hexacyanoferrate was electrochemically characterized using the CV technique, which is a powerful method to study adsorption processes. The electrocatalytic activities of ferrocene and its derivatives in electrochemistry are well known for producing redox peaks with high reversibility mainly due to the oxidation of PB [26]. The incorporation of PB into the nanocomposite (PB@CNT-PPy) was observed by distinguishing redox pairs of cathodic and anodic peaks at −0.38 V and close to 0.6 V, respectively, due to the reversible oxidation of PB, a shown in curve IV of Figure 2a. The curves of CV were registered in the presence of KCl at 0.1 V. s^−1^ scan rate; redox peaks were not observed in the films formed only by PPy and PPy-CNT, although the polymer pyrrole films can produce Faradaic currents due to anionic charges on the surface, resulting from their synthesis [27,28].

Surface-confined PB on the nanocomposite was also studied using CV by varying the scan rate from 10 to 110 mV. s^−1^ with a potential window from −0.8 to 1.0 V, in the presence of the support electrolyte of 0.1 mol L^−1^ KCl. The increase of redox peaks increased with an increase in scan rates, and linear curves for the current of anodic (Ipa) and cathodic (Ipc) peaks versus the scan rates were observed. Correlation coefficients were calculated for Ipa (r = 0.993) and Ipc (r = 0.988), proving that the PB was successfully confined to the electrode surface of the PPy-CNT nanocomposite (Figure 2b and inset). Additionally, the slope of the curves plotted by the logarithm scan rate versus the logarithm of the cathodic and anodic peaks were 0.538 and 0.588, respectively, indicating that the PB-confined generated electroactive species, resulting in a diffusion-controlled process since the obtained values were very close to the theoretical value of 0.5, which is expressed by an ideal reaction of the diffusion-controlled electrode process (Figure 2c). Hence, incorporation of PB in nanocomposites was successfully achieved, probably attributed to the cycling of hexacyanoferrate that was induced by the addition of Fe^3+^ ions [26].

Electrochemical stability of the PB@CNT-PPy/SPE redox peaks was investigated by subjecting the nanocomposite film to 20 successive CVs. The relative standard deviation (RSD%) for anodic and cathodic peaks (Ipa and Ipc) were 3.4 and 3.1%, respectively, indicating good stability (RSD < 5%). PB is an inorganic complex salt containing two differently charged iron ions and negatively charged hexacyanoferrate ions [Fe(CN)_6_]^4−^ [29].

The PB amount deposited on the PPy is dependent on the compound used in the Prussian white to Prussian blue reaction conversion according to the reaction [30]:Fe^2+^ + [Fe(CN)_6_]^3−^→ Fe_3_[Fe(CN)_6_]^2^(PB) (2)

Previous studies have demonstrated integrating monomers derived from conductive polymers acting as a reducing agent in converting Fe^3+^ to Fe^+2^ [31]. Yang et al. (2017) [32] stated that monomers of Py contributed to the generation of Fe^2+^ ions in the formation of PPy, following this reaction:Py + Fe^3+^ →PPy + Fe^2+^(3)

Fe^2+^ ions freely react with the hexacyanoferrate ions and enhance the deposition rates of PB. Herein, the synthesis of PB@CNT-PPy resulted in the efficient integration of the PB into the nanocomposite, improving the electrical proprieties.

### 3.2. Chemical Characterization of the PB@CNT-PPy Nanocomposite Film

The FTIR analysis was used for chemical characterization of the PB@CNT-PPy film and its controls. The spectrum of the bare SPE was observed in the presence of the C = O stretching vibrations (1727 cm^−1^), C = C from sp^2^ bonds (1550–1450 cm^−1^) and C-O vibrations (1270 cm^−1^) (Figure 3a). The profile obtained was attributed to mercaptan enriched carbon ink composition revealing the presence of carbonyl, carboxyl and epoxy groups [33]. FTIR spectrum from the SPE modified with PPy/CNT (Figure 3b) shows a broad peak at 3070 cm^−1^, which is characteristic of the O-H stretch of hydroxyl groups. The peaks appeared at 2896, 1420 and 1200 cm^−1^ corresponds to C-H, C = O and C-O stretching, respectively, exhibiting typical bands of the carboxylic groups derived from the CNT [34]. In addition, at peaks 1566 and 1460 cm^−1^, C = C backbone stretching and C-N stretching vibration present in the PPy ring were observed, respectively [35]. The IR spectrum of the PB@CNT-PPy film assembled on the SPE (Figure 3c) shows a strong peak at 2057 cm^−1^, which was assigned as stretching vibration of the C ≡ N group in potassium hexaferricyanide [36]. This FTIR spectrum, as compared to spectra controls, confirms the presence of the PB chemical mediator in the nanocomposite.

### 3.3. Electrochemical Characterization of the ZIKV Immunosensor

Stepwise modifications on the electrode surface were characterized by cyclic voltammetry (CV) in the presence of the KCl (0.1 mol L^−1^) solution (Figure 4). The bare SPE showed an absence of remarkable redox peaks (curve I). After assembling the PB@CNT-PPy film on bare SPE, the anodic and cathodic peak current at 0.6 V and −0.1 V (vs. Ag/AgCl _KCl sat_), respectively, was observed (curve II). A significant increase in the current signals was observed after electrosynthesis of the PB@CNT-PPy film. This is attributed to the redox activity of the nanocomposite film and the excellent conductivity, promoting an increase in electron transfer. After anti-NS2B immobilization, a decrease in the current value was observed, indicating that the electron transfer was reduced due to the insulating feature of the biomolecules (curve III). The electron transfer on the sensor surface was measured by changes in the oxidative states of the PB on the interface electrolyte/sensor, which intrinsically originated from the charge transfer between Fe^3+^ and Fe^2+^ in the nanocomposite film. A further small decrease in the redox peak values was observed after the blocking of the remaining active with glycine (curve IV).

### 3.4. Immunosensor Response to NS2B Protein

The analytical performance of the immunosensor for detection of antibody–antigen reaction depends on the amount of immobilized antibodies on the sensor surface. Therefore, the optimal amount of anti-NS2B immobilized on PB@CNT-PPy/SPE was obtained through successive antibody incubation with one aliquot (5 µL) of anti-NS2B concentration at 5 mg L^−1^. The optimal anti-NS2B concentration in the range of 0.0005 to 0.004 g L^−1^ was reached at 0.4 mg L^−1^, which is associated with the non-free binding group present in the PB@CNT-PPy film (Figure 5).

The immunosensor was subjected to a successive response to NS2B native protein expressed by Vero culture supernatant collected on the seventh day of ZIKV inoculation. In this sense, SPE was incubated for 15 min in a moist chamber at 4 °C with 5 µL of ZIKV isolates in cell-culture supernatant (7th day) with approximately a titer of 2.2 × 10^6^ PFU mL^−1^ diluted 1:4 in PBS solution (0.01 mol L^−1^, pH 7.4). As a control, negative cell-culture supernatant (without ZIKV inoculation) was used. Analytical responses exhibited the relationship between the current variations (ΔIpa%) and the number of successive incubations with cell-culture supernatant with inoculated ZIKV (Figure 6a). The current responses demonstrated that the immunosensor was able to discriminate NS2B native in complex samples of positive and negative samples of cell-culture supernatants infected with ZIKV, showing specificity on responses. The linear regression equation of the analytical response for positive samples showed a good a correlation coefficient of 0.986 (*p* < 0.0001, *n* = 6) and a low relative error (<<1%) (inset: Figure 6a). Electrogenerated chemiluminescence [37] and surface-enhanced Raman scattering (SERS) [38] {Formatting Citation} immunoassays have been described in the literature and have been proposed for ZIKV detection using antibodies against E and NS1 ZIKV proteins, respectively. More recently, Mishra et al. described a sensitive ELISA based on the immunoreactivity of NS2B protein [14]. This study demonstrated an increase in the sensitivity for ZIKV detection of 98% in the early-convalescent phase and also suggests non-cross-reactivity with the dengue virus. Our results suggest a novel method of serologic testing for ZIKV diagnostics based on NS2B detection by using an electrochemical immunosensor compatible with point-of-care sensor technology.

The performance of the immunosensor was also evaluated in real samples through incubations with the serum of the ZIKV patients, diluted at 1:2 in PBS. After incubations, the surface electrodes were gently washed, and the measurements were performed in KCl (0.1 mol L^−1^). The analytical curve was obtained by chronoamperometric measurements with current values registered at 2 s after applying 0.4 V for 10 s. Response time after 2 s was established according to the greatest difference between amperometric responses. As shown in Figure 6b, this immunosensor exhibited a linear relationship between the current values and the successive injections of the positive ZIKV serum samples. The correlation coefficient calculated was 0.997 (*p* < 0.0001, *n* = 6) and a low relative error (<<1%), exhibiting a strong relationship found between the chronoamperometric measurements and successive injections of the positive ZIKV serum samples. The selectivity of the ZIKV immunosensor was evaluated by incubating different electrodes with human samples: +ZIKV/−DENV and −ZIKV/+DENV, both diluted in 1:2; 1:4; 1:8; 1:16, and 1:32. The overlapping of the relative current obtained for positive and negative samples is shown in Figure 6c. The results showed a proportional decrease of the current responses with the increase of the dilution factor for the positive serum sample when compared to the negative. The current response revealed significant differences from positive and negative below 1:4 PBS diluted. These results show that this ZIKV immunosensor has good selectivity and was efficient in differentiating the positive and negative serum samples in complex matrices of infected patients with ZIKV, considering negative samples for ZIKV infected of DENV. This approach shows the potential for a POCT tool, allowing for easier epidemiologic control since the response is obtained by a practical analytical method.

## 4. Conclusions

A point-of-care immunosensor based on a low-cost and practical screen-printed electrodes technology was successful developed for NS2B detection, a potential marker of ZIKV infection. The synergetic effect of the PB@CNT-PPy nanocomposite enabled probeless electrochemical detection without added redox probe solutions, simplifying the detection method. The immunosensor exhibited a good selectivity, distinguishing between negative from positive ZIKV serum, even if DENV was present. This immunosensor represents a potential alternative for the detection of ZIKV infection as a point-of care diagnostic tool.

## Figures and Tables

**Figure 1 biosensors-11-00157-f001:**
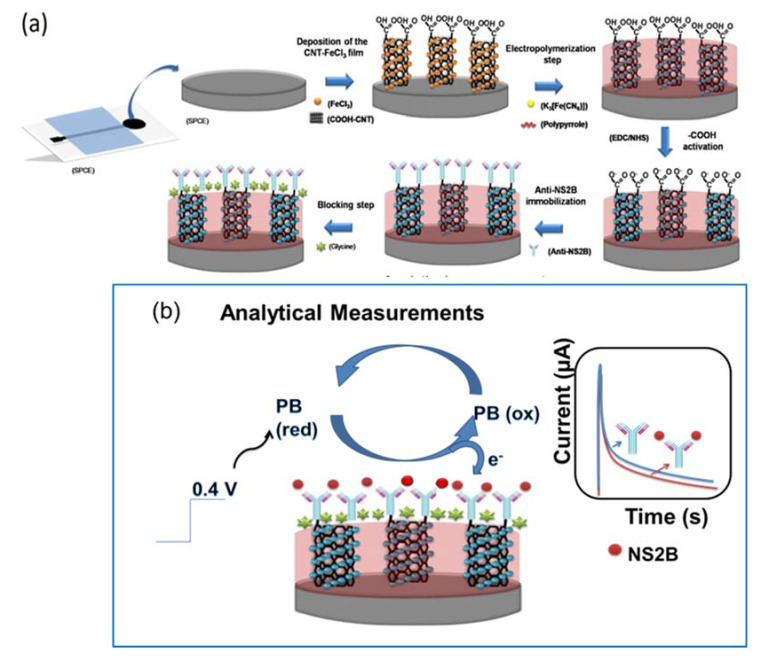
Schematic diagram of ZIKV immunosensor. (**a**) Preparation steps of the electrode and (**b**) analytical measurement principle.

**Figure 2 biosensors-11-00157-f002:**
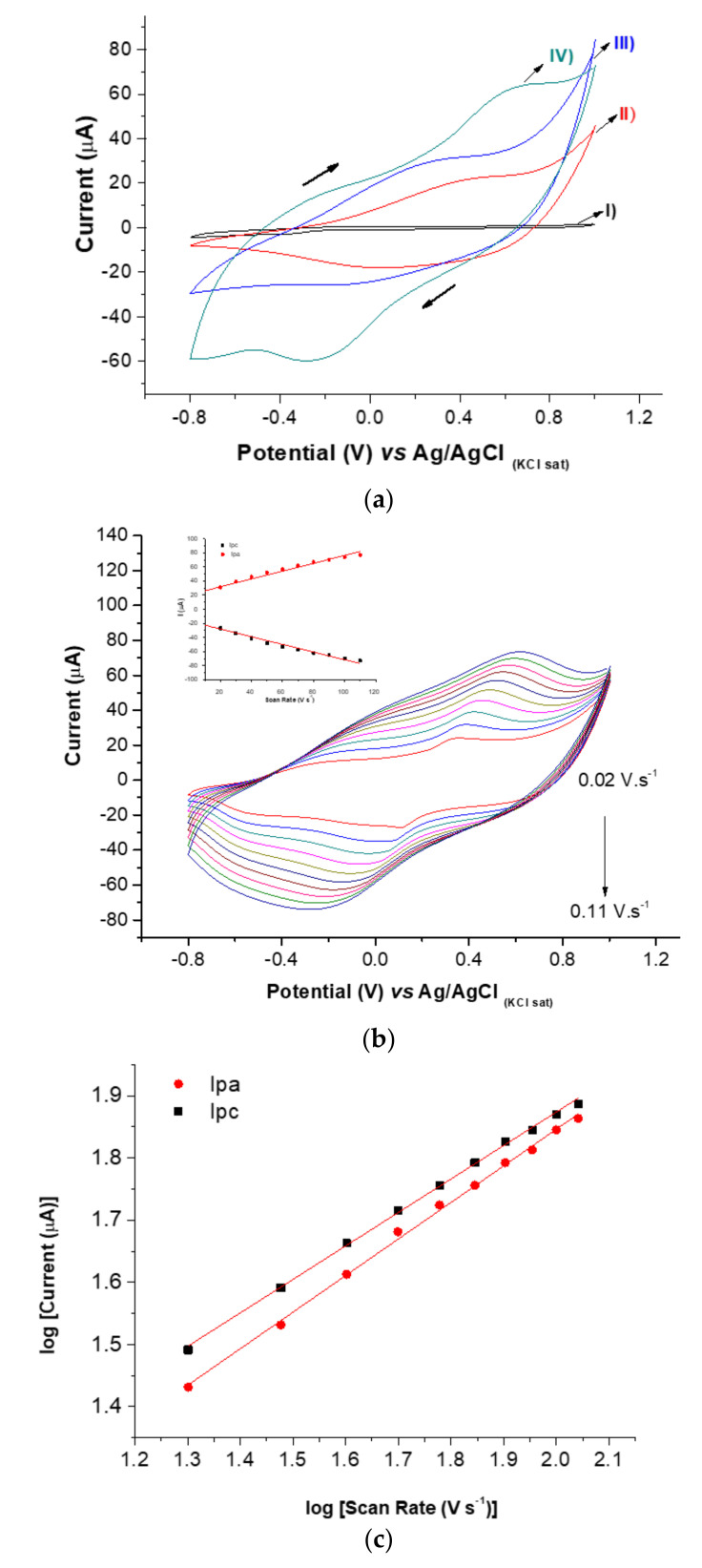
(**a**) Cyclic voltammograms at 0.1 V s^−1^ scan rate: (I) bare SPE; (II) PPy/SPE; (III) PPy-CNT and (IV) PB@CNT-PPy. (**b**) Effect of the scan rate on the current response of the PB@CNT-PPy/SPE (from 0.01 to 0.12 V s^−1^) (inset: relationship between Ipa and Ipc vs. scan rate). (**c**) Plot of log Ipa and Ipc vs. log of scan rate. All measurements were performed in KCl (0.1 mol L^−1^) as support electrolyte.

**Figure 3 biosensors-11-00157-f003:**
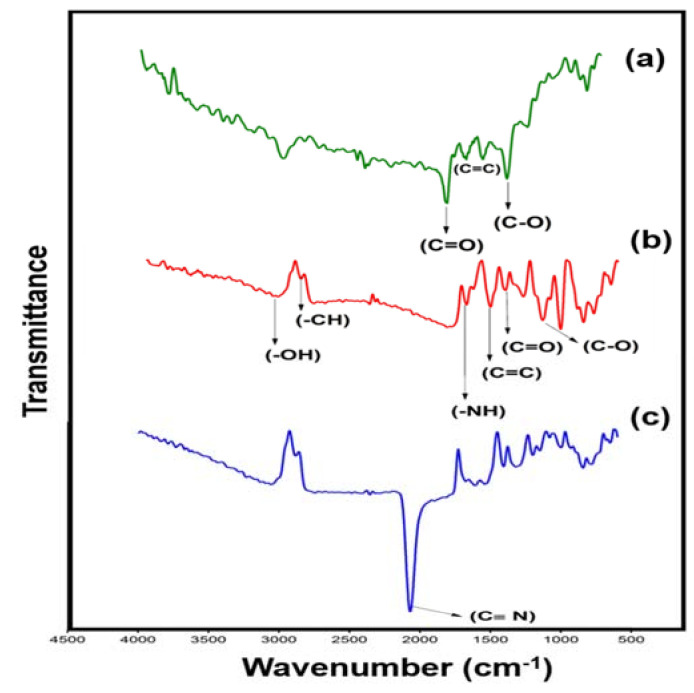
FTIR spectrum of the (**a**) bare SPE, (**b**) PPy/CNT/SPE and (**c**) PB@CNT-PPy/SPE.

**Figure 4 biosensors-11-00157-f004:**
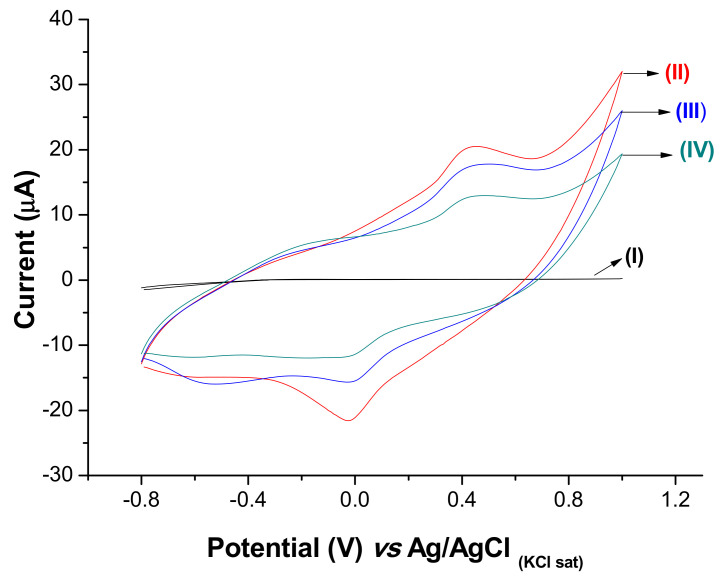
Cyclic voltammograms of the stepwise preparation of the immunosensor: (I) bare SPE, (II) PB@CNT-PPy/SPE, (III) anti-NS2B/PB@CNT-PPy/SPE and (IV) Gly/anti-NS2B/PB@CNT-PPy/SPE. Measurements were performed in the presence of KCl (0.1 mol L^−1^) at a 0.05 V s^−1^ scan rate.

**Figure 5 biosensors-11-00157-f005:**
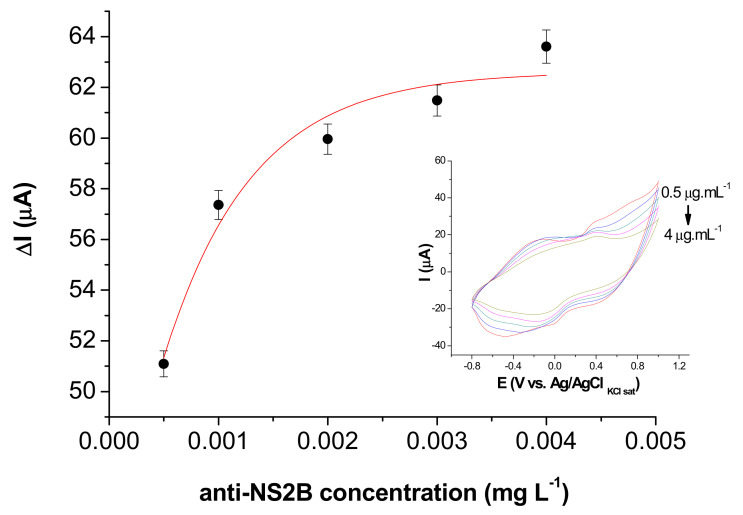
Optimization of anti-NS2B concentrations (0.5; 0.1; 2; 3 and 4 mg L^−1^) immobilized (inset: CVs of the immunosensor in the different anti-NS2B concentrations). Measurements were performed in KCl (0.1 mol L^−1^) solution at a 0.05 V s^−1^ scan rate.

**Figure 6 biosensors-11-00157-f006:**
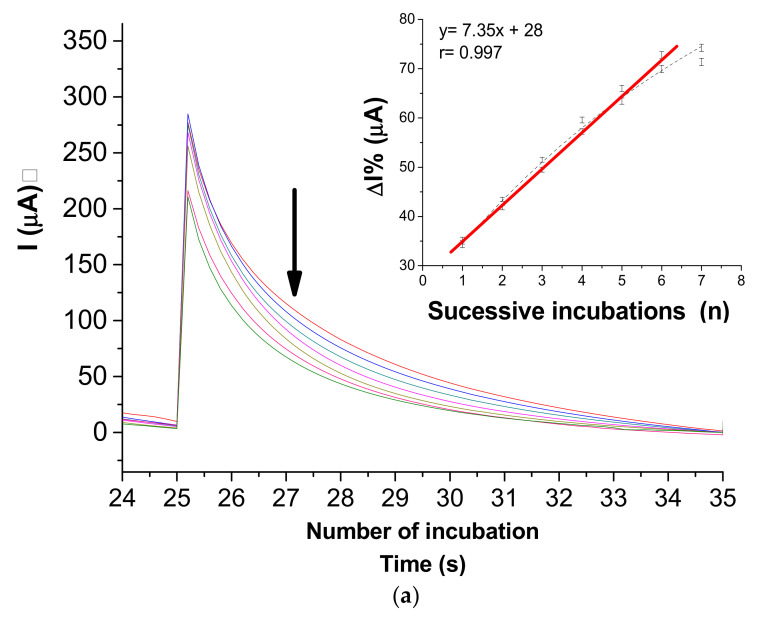
(**a**) Chronoamperograms obtained after successive incubations with supernatants of the infected ZIKV cell-culture and PBS washes before measurements obtained in presence 0.1 mol L^−1^ KCl solution at 0.4 V fixed potential. Inset: linear fit adjusted for successive cell-culture incubations. (**b**) Analytical curve of the immunosensor in response to successive incubations with serum samples 1:100 PBS dilution of the ZIKV (positive)/DENV (negative) and ZIKV (negative)/DENV (positive), measured by chronoamperograms at 0.4 V vs. Ag/AgCl. (**c**) Bar diagram of current (with error bar of three replicates) as function of incubation number of ZIKV (positive)/DENV (negative) and ZIKV (negative)/DENV (positive).

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
