# Peer review of "A Label and Probe-Free Zika Virus Immunosensor Prussian Blue@carbon Nanotube-Based for Amperometric Detection of the NS2B Protein"

_biosensors, 2021, doi:10.3390/bios11050157_

Round 1
Reviewer 1 Report
This manuscript introduces an amperometric immunosensor for ZIKA virus which shows good potential for point of care detects.
- Figure 2: x-axis, please change 0,2 to 0.2 and follow the same for the rest of the coordinates.
- Figure 4: what does line (V) refers to. It is missing in the caption.
- Figure 5: for DI% in the inset, there is no need to put a unit. What it the ZIKV number here? How does this number correlates to the pfu value? For calibration, it is important to actually correlate with the actual viral number and also cover the clinical relevant range of viral concentration. Similarly for Figure 6, please clarify the ZIKV number.
- Figure 6: can we just use the serum dilution factor rather than ln(serum dilution factor)?
- Please check through the manuscript carefully. There are quite a number of typo errors.
Author Response
This manuscript introduces an amperometric immunosensor for ZIKA virus which shows good potential for point of care detects.
- Figure 2: x-axis, please change 0,2 to 0.2 and follow the same for the rest of the coordinates.
We thank. All figures have been revised.
- Figure 4: what does line (V) refers to. It is missing in the caption.
We appreciate the comments. There are a mistake and the curve was curve V in Figure 4 has been removed.
- Figure 5: for DI% in the inset, there is no need to put a unit. What it the ZIKV number here? How does this number correlates to the pfu value? For calibration, it is important to actually correlate with the actual viral number and also cover the clinical relevant range of viral concentration. Similarly for Figure 6, please clarify the ZIKV number.
The information of the PFU title used in the assay was added in the text in the line 297.
We agree with the review, there is not have unit as %. It is corrected.
- Figure 6: can we just use the serum dilution factor rather than ln(serum dilution factor)?
We agree with the observations. The scale of the Figure 6(b) has been corrected.
- Please check through the manuscript carefully. There are quite a number of typo errors.
The manuscript was all revised.
#Review 2
Manuscript presents the amperometric immunosensor for the detection of NS2B protein present in Zika virus infected patients. The sensor developed by authors uses screen printed electrode with Prussian Blue / carbon nanotube / polypyrrole nanocomposite. It has good selectivity, allowing the detection of Zika virus in patient's serum, without the risk of misdiagnosis due to the interferences from Dengue virus.
Authors clearly presented the preparation of the electrode (including materials used) as well as the electrochemical analysis of this electrode using voltammetry and infrared spectroscopy. The difference in the response towards Zika-positive and Dengue-positive samples was also presented.
Nevertheless, some issues must be addressed before the publication, including:
- In line 332, authors state that they use “neperian logarithm”. This is not true! They use natural logarithm. The difference between Napierian logarithm and natural logarithm is well described e.g. in the publication of D. Roegel (see https://hal.inria.fr/inria-00543934/document)
We appreciate the observation. The text and legend of the Figure 6 was changed to the sample dilution values.
- The x-axis of figure 7 is correctly marked as “Ln(serum dilution factor)”, while in figure caption there is “Ln incubation number”. Please use the dilution factors under each pair of bars instead of continuous logarithmic scale. (Current form of presentation suggests that e.g. 1:4-dillution bars are made at -1.25 for ZikV and -1.5 for DenV.)
The observations have been changed in the text and in the Figure.
- The current redox peaks vs. square root of scan rates (mentioned in lines 204 and 213) should be shown.
The figure was adjusted as recommended.
- What is the curve (V) in Figure 4? It is not described neither in the text nor the figure caption.
We appreciate the comments. There was a mistake and the curve V in Figure 4 was removed.
- Figure S1 (unavailable for the reviewers) should be included in the manuscript.
Figure S1 was as Figure 5 and all other figures in the text renumbered.
Some other minor issues - mostly editorial, that should be addressed include:
- The chemical formulas and units should be written properly throughout the manuscript {e.g. in lines 144-145 should be “FeCl3∙H2O (0.2 mol L-1)” instead of “FeCl3.H2O (0.2 mol.L-1)”; “3 µL” not “3µL”; and “1 g L-1” instead of “1 mg.mL-1”.}
We appreciate the observation. All the text was revised.
- Figure 1 should be larger and in higher resolution
The resolution of the Figure 1 was corrected.
- In line 181 there is “Pb-PPy@CNT” and in line 182 “Pb@PPy-CNT”. This should be unified throughout the manuscript
All the text was revised.
- Figure 3 seems to be horizontally stretched, and as a result looks visually bad
We send the figure in a file.
- Figure 5 is not mentioned in the text. It should be
The Figure 5 was mentioned in the line 286.
- Reviewer recommends a language check of the whole manuscript (see e.g. lines 20-23, 34-36, 57-59, 112-113, etc.)
All the manuscript was revised, as requested.
Reviewer 2 Report
Manuscript presents the amperometric immunosensor for the detection of NS2B protein present in Zika virus infected patients. The sensor developed by authors uses screen printed electrode with Prussian Blue / carbon nanotube / polypyrrole nanocomposite. It has good selectivity, allowing the detection of Zika virus in patient's serum, without the risk of misdiagnosis due to the interferences from Dengue virus.
Authors clearly presented the preparation of the electrode (including materials used) as well as the electrochemical analysis of this electrode using voltammetry and infrared spectroscopy. The difference in the response towards Zika-positive and Dengue-positive samples was also presented.
Nevertheless, some issues must be addressed before the publication, including:
- In line 332, authors state that they use “neperian logarithm”. This is not true! They use natural logarithm. The difference between Napierian logarithm and natural logarithm is well described e.g. in the publication of D. Roegel (see https://hal.inria.fr/inria-00543934/document)
- The x-axis of figure 7 is correctly marked as “Ln(serum dilution factor)”, while in figure caption there is “Ln incubation number”. Please use the dilution factors under each pair of bars instead of continuous logarithmic scale. (Current form of presentation suggests that e.g. 1:4-dillution bars are made at -1.25 for ZikV and -1.5 for DenV.)
- The current redox peaks vs. square root of scan rates (mentioned in lines 204 and 213) should be shown.
- What is the curve (V) in Figure 4? It is not described neither in the text nor the figure caption.
- Figure S1 (unavailable for the reviewers) should be included in the manuscript.
Some other minor issues - mostly editorial, that should be addressed include:
- The chemical formulas and units should be written properly throughout the manuscript {e.g. in lines 144-145 should be “FeCl3∙H2O (0.2 mol L-1)” instead of “FeCl3.H2O (0.2 mol.L-1)”; “3 µL” not “3µL”; and “1 g L-1” instead of “1 mg.mL-1”.}
- Figure 1 should be larger and in higher resolution
- In line 181 there is “Pb-PPy@CNT” and in line 182 “Pb@PPy-CNT”. This should be unified throughout the manuscript
- Figure 3 seems to be horizontally stretched, and as a result looks visually bad
- Figure 5 is not mentioned in the text. It should be
- Reviewer recommends a language check of the whole manuscript (see e.g. lines 20-23, 34-36, 57-59, 112-113, etc.)
Author Response
#Review 2
Manuscript presents the amperometric immunosensor for the detection of NS2B protein present in Zika virus infected patients. The sensor developed by authors uses screen printed electrode with Prussian Blue / carbon nanotube / polypyrrole nanocomposite. It has good selectivity, allowing the detection of Zika virus in patient's serum, without the risk of misdiagnosis due to the interferences from Dengue virus.
Authors clearly presented the preparation of the electrode (including materials used) as well as the electrochemical analysis of this electrode using voltammetry and infrared spectroscopy. The difference in the response towards Zika-positive and Dengue-positive samples was also presented.
Nevertheless, some issues must be addressed before the publication, including:
- In line 332, authors state that they use “neperian logarithm”. This is not true! They use natural logarithm. The difference between Napierian logarithm and natural logarithm is well described e.g. in the publication of D. Roegel (see https://hal.inria.fr/inria-00543934/document)
We appreciate the observation. The text and legend of the Figure 6 was changed to the sample dilution values.
- The x-axis of figure 7 is correctly marked as “Ln(serum dilution factor)”, while in figure caption there is “Ln incubation number”. Please use the dilution factors under each pair of bars instead of continuous logarithmic scale. (Current form of presentation suggests that e.g. 1:4-dillution bars are made at -1.25 for ZikV and -1.5 for DenV.)
The observations have been changed in the text and in the Figure.
- The current redox peaks vs. square root of scan rates (mentioned in lines 204 and 213) should be shown.
The figure was adjusted as recommended.
- What is the curve (V) in Figure 4? It is not described neither in the text nor the figure caption.
We appreciate the comments. There was a mistake and the curve V in Figure 4 was removed.
- Figure S1 (unavailable for the reviewers) should be included in the manuscript.
Figure S1 was as Figure 5 and all other figures in the text renumbered.
Some other minor issues - mostly editorial, that should be addressed include:
- The chemical formulas and units should be written properly throughout the manuscript {e.g. in lines 144-145 should be “FeCl3∙H2O (0.2 mol L-1)” instead of “FeCl3.H2O (0.2 mol.L-1)”; “3 µL” not “3µL”; and “1 g L-1” instead of “1 mg.mL-1”.}
We appreciate the observation. All the text was revised.
- Figure 1 should be larger and in higher resolution
The resolution of the Figure 1 was corrected.
- In line 181 there is “Pb-PPy@CNT” and in line 182 “Pb@PPy-CNT”. This should be unified throughout the manuscript
All the text was revised.
- Figure 3 seems to be horizontally stretched, and as a result looks visually bad
We send the figure in a file.
- Figure 5 is not mentioned in the text. It should be
The Figure 5 was mentioned in the line 286.
- Reviewer recommends a language check of the whole manuscript (see e.g. lines 20-23, 34-36, 57-59, 112-113, etc.)
All the manuscript was revised, as requested.
Round 2
Reviewer 1 Report
The manuscript has been improved.
Author Response
thanks